# NMR spectroscopy-based analysis of gallstones of cancerous and benign gallbladders from different geographical regions of the Indian subcontinent

Mohd Adnan Siddiqui[1,2], Navneet Dwivedi[2], Mohammed Haris Siddiqui[1], S. V. Rana[3], Anil Sharma[4], N. R. Dash[5], Rebala Pradeep[6], Ranjit Vijayahari[7], Anu Behari[8], V. K. Kapoor[9], Neeraj Sinha[2]*

1 Department of Bioengineering, Integral University, Lucknow, India, 2 Centre of Biomedical Research, SGPGIMS Campus, Lucknow, India, 3 Department of Gastroenterology, PGIMER, Chandigarh, India, 4 Vivekanand Medical Institute, Palampur, Kangra, Himachal Pradesh, India, 5 All India Institute of Medical Sciences (AIIMS), New Delhi, India, 6 Asian Institute of Gastroenterology, Hyderabad, India, 7 Department of GI Surgery, Cosmopolitan Hospital, Thiruvananthapuram, Kerala, India, 8 Department of Surgical Gastroenterology, SGPGIMS, Lucknow, India, 9 Mahatma Gandhi Medical College & Hospital, Jaipur, Rajasthan, India

* neerajcbmr@gmail.com, neeraj.sinha@cbmr.res.in

**Data Availability Statement:** The minimal data set has been deposited in the Zenedo repository at the

## Abstract

Analysis of the chemical composition of gallstones is vital for the etiopathogenesis of gallstone diseases that can ultimately help in the prevention of its formation. In the present study, gallstones from seven different regions of India were analyzed to highlight the major difference in their composition. Also, gallstones of different pathological conditions i.e., benign (chronic cholecystitis, CC) and malignant gallbladder disease (gallbladder cancer GBC) were characterized. The type of polymorphs of cholesterol molecules was also studied to provide insight into the structure of gallstones. $^1$H solution state NMR spectroscopy 1D experiments were performed on a total of 94 gallstone (GS) samples collected from seven different geographical regions of India. Solid-State NMR spectroscopy $^{13}$C cross-polarization magic angle spinning (CPMAS) experiments were done on the 20 CC GS samples and 20 GBC GS samples of two regions. $^1$H NMR spectra from the solution state NMR of all the stones reveal that cholesterol was a major component of the maximum stones of the north India region while in south Indian regions, GS had very less cholesterol. $^{13}$C CPMAS experiments reveal that the quantity of cholesterol was significantly more in the GS of CC in the Lucknow region compared with GBC stones of Lucknow and Chandigarh. Our study also revealed that GS of the Lucknow region of both malignant and benign gallbladder diseases belong to the monohydrate crystalline form of cholesterol while GS of Chandigarh region of both malignant and benign gallbladder diseases exists in both monohydrate crystalline form with the amorphous type and anhydrous form. Gallstones have a complicated and poorly understood etiology. Therefore, it is important to understand the composition of gallstones, which can be found in various forms and clinical conditions. Variations in dietary practices, environmental conditions, and genetic factors may influence and contribute to the

following link: (https://zenodo.org/record/8000849).

**Funding:** The authors received no specific funding for this work.

**Competing interests:** The authors have declared that no competing interests exist.

formation of GS. Prevention of gallstone formation may help in decreasing the cases of gallbladder cancer.

## Introduction

Gallstone disease is one of the major causes of gastrointestinal conditions that bring about abdominal pain condition associated with hospitalization and surgical intervention. It continues to be a serious health issue and financial burden that impacts millions of people globally [1]. In the Indian population, gallstone disease affects about 6% of people [2]. Gallstones (GS) and gallbladder cancer frequently coexist; patients with a history of gallstones that measure more than 3.0 cm in diameter are said to be more likely to develop gallbladder cancer [3]. Despite being an uncommon disease, India accounts for close to 10% of the world's cases of gallbladder cancer. The northern region of India has a higher prevalence of GBC than the southern region [4].

The liver produces and secretes bile, a physiological aqueous solution that is largely necessary for emulsifying lipids before digestion. Cholesterol, conjugated bilirubin, bile salts, phospholipids, electrolytes, and water make up the majority of bile. Bile enters the gallbladder via the hepatic duct, where it is stored and concentrated through trans-mucosal absorption. To aid in the breakdown and absorption of lipids during digestion, the gallbladder secretes bile into the small intestine. The agglomeration of the various bile contents leads to cholelithiasis or the formation of gallstones. Due to numerous disease conditions, the chemical components of bile become disrupted, changing its solubility [5]. Gallstone development is triggered by the supersaturation of bile cholesterol and a defective bilirubin conjugation reaction [6]. Some other constituents of the bile, like magnesium salts and calcium salts, combine with cholesterol to form the gallstone. There are no effective medications on the market right now to treat gallstone diseases. The standard treatment for gallstone disease is cholecystectomy or surgery to remove the gallbladder.

Gallstones can be amorphous or crystalline, with varied colors, including black, brown, yellow, and white, varying diameters ranging from a few millimeters to six centimeters, and distinct geometries [7]. Gallstones are primarily divided into three categories: cholesterol stones, mixed stones, and pigment stones [8]. The typical colors of cholesterol stones are creamy-white, yellow, brown, or greenish. Mixed stones are mostly of dark brown and brownish-yellow color. The majority of pigment stones are black and brown. Several studies have revealed the variation in the type and composition of gallstones from different ethnic populations. Many risk factors such as age, gender, obesity, pregnancy, oral contraceptive therapy, diabetes mellitus, serum lipids levels, cirrhosis, family history, and genetics have also been identified to be associated with gallstone formation [9, 10]. It is also reported that in 80% of the Indian population with GBC, gallstones are present [4]. Other cofactors associated with GBC incidence are older age, chronic *Salmonella typhi* infection, *Helicobacter pylori* infection, heavy metals, chemicals, exposure to pollutants, adulterated mustard oil, smoking, and lower socio-economic status [10].

The pathogenesis of gallstones is complicated, and its existence in wide variations is not well understood. The classification of stones based on morphological appearance solely may be misleading, so the analysis of the chemical composition of gallstones must be conducted for a more precise and accurate classification. The chemical constituents of gallstones must be identified in order to elucidate the etiopathogenic factors, which may then be used to implement treatment and prevention plans. Previously some studies have been performed for

analysing the compositions of gallstones from Lucknow, Meerut, Hyderabad, and Chennai [11, 12]. The purpose of the current study was to investigate the incidence of GS disease among people from seven different geographical areas of India by analyzing gallstones. Various techniques have been employed for the characterization of gallstone diseases in the past. X-ray diffraction [13], Fourier-transform infrared spectroscopy (FTIR) [11, 14, 15], particle-induced X-ray emission [16], scanning electron microscopy (SEM) [17, 18], magnetic resonance imaging (MRI) [19] and nuclear magnetic resonance (NMR) spectroscopy [11, 17, 20–23], have been used extensively to study the composition of gallstones and provide significant information to the scientific community. But still, we are nowhere close to the prevention of gallstones. As we are getting modern with time, people from the urban cities of India are shifting more towards the Western diet. Diet is always considered to be one of the main factors for GS occurrence [24, 25]. So, from time to time, research on the composition of gallstones including in several parts of India is required to highlight whether with time trends there is any change in the compositional pattern of gallstones or not. India is such a vast country, ranging from the Himalayas in the north to Kanyakumari in the south, so in the present study, we have analyzed variation in the level of constituents of gallstones across the seven different regions of India. The study included the gallstones from the Himalayan valley Kangra, which lies in the north-western region of India to the southernmost district of India, i.e., Thiruvananthapuram, Kerala. In this way, as per our knowledge, this is the first time the gallstones have been analyzed covering most of the parts (seven regions) of India (Fig 1).

The applicability of NMR spectroscopy to provide insight from the samples of different disease conditions is significant. It is non-invasive, highly reproducible, does not need extensive sample preparation, and is quite rapid. In this study, we have utilized the potential of solution state NMR spectroscopy to analyze gallstones from different geographical regions in India. Also, gallstones of benign gallbladder disease (chronic cholecystitis CC) and malignant gallbladder disease (gallbladder cancer GBC) were characterized using solid state NMR spectroscopy. Gallstone structural information is provided by $^{13}$C CPMAS for both CC and GBC related disorders. This is the first time that the combination of applicability of both solution state NMR spectroscopy and solid state NMR spectroscopy has been adapted to characterize the gallstones from different geographical regions and different pathological conditions, i.e., CC stones & GBC stones. Such a study will be necessary to get meaningful insight into the composition of gallstones present in different aetiological conditions of gallstone diseases from different geographical conditions. Identifying the chemical composition and mechanism of the formation of gallstones is important in delineating the etiopathogenic factors, which is sequentially helpful in executing therapeutic and preventive strategies.

## Materials and methods

### Collection of gallstones

In the current study, human gallstones from seven regions, Chandigarh, Delhi, Kangra, Lucknow, and West Bengal (grouped as North India), and from Hyderabad and Thiruvananthapuram (grouped as South India) were analyzed. GS samples were obtained after cholecystectomy of 94 patients of cholelithiasis and CC only (no GBC). GS samples were provided by the Department of Gastroenterology, Postgraduate Institute of Medical Education and Research (PGIMER), Chandigarh (n = 20), Department of Surgical Gastroenterology, Sanjay Gandhi Postgraduate Institute of Medical Sciences (SGPGIMS), Lucknow (n = 19), Vivekanand Medical Institute, Palampur, Kangra (n = 11), All India Institute of Medical Sciences (AIIMS), New Delhi (n = 9), Asian Institute of Gastroenterology (AIG), Hyderabad (n = 11) and Department of GI Surgery, Cosmopolitan Hospital, Thiruvananthapuram

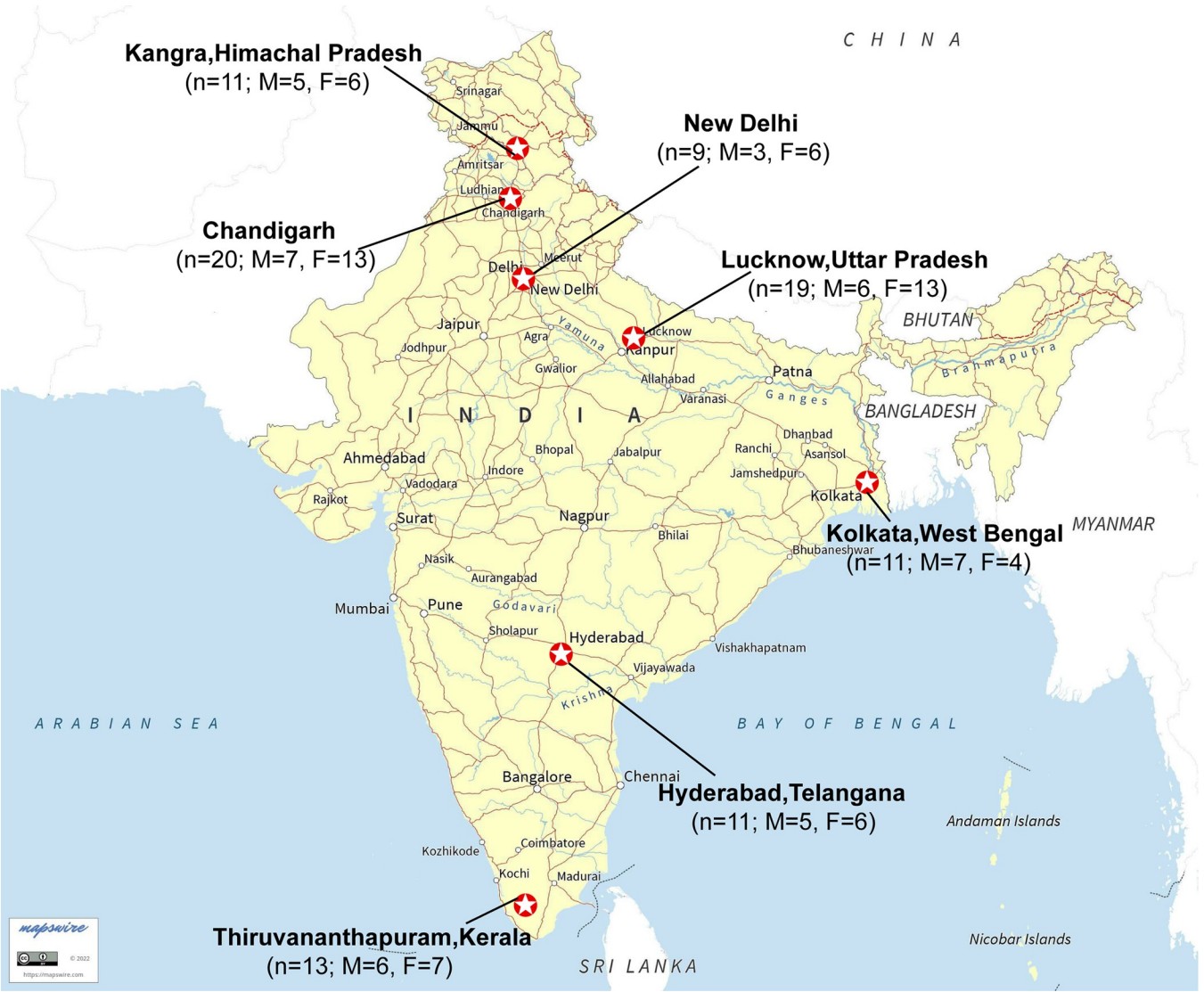

**Fig 1. Indian map showing the location of collected samples, number of samples (n) from each region, and sexuality (M = No. of male patients, F = No. of female patients) of gallstone disease patients included in the study.** The map was created by the authors using basemap from mapswire (https://mapswire.com/).

(n = 13). GS from patients of West Bengal origin (n = 11) were provided by a surgeon at the Asian Institute of Gastroenterology, Hyderabad.

In addition, GS of GBC patients were obtained from the Department of Gastroenterology, PGIMER, Chandigarh (n = 10) and Department of Surgical Gastroenterology, SGPGIMS, Lucknow (n = 10) to perform the comparative analysis for the type of cholesterol molecules with CC stones of PGIMER, Chandigarh (n = 10), and SGPGIMS, Lucknow (n = 10). The institutional ethical committee of SGPGIMS, Lucknow approved this study. All the CC and GBC GS samples have been collected over a period of 13 months from July 2021 to July 2022. The written informed consent form was obtained from all the patients included in the study. To completely remove all traces of bile from the collected GS, various pre-treatments are required before the experiments can begin. Gallstones were cleaned with double-distilled, deionized water, and all of them were kept for sunlight drying. Gallstones' internal and

external physical traits were noted. Gallstones were carefully crushed to a fine powder using a pestle and mortar. Until the experiments were carried out, the powdered gallstones were kept in airtight vials at room temperature. When multiple stones were present in a single patient, only one stone was used for analysis. Deuterated chloroform ($CDCl_3$) was used as a solvent in all the solution NMR experiments and trimethylsilyl propionic acid-d4 (TSP) is used as an internal reference. All solvents and chemicals used were of Sigma-Aldrich (St Louis, MO, USA).

## NMR experiments

**Solution state NMR experiments.** The powdered GS (~10mg) was taken in an eppendorf and dissolved properly in deuterated chloroform (500 μl). The dissolved gallstones solution was taken in a separate 5-mm NMR tube. The NMR tubes used were from Wilmad Glass, USA. A reusable, sealed capillary containing TSP dissolved in $D_2O$ (0.5mM) was placed inside the 5-mm NMR tube. It is used for chemical shift and quantitation reference. All the NMR experiments were performed at 298 K on a Bruker Biospin Avance III 800 MHz NMR (Bruker GmBH, Germany) spectrometer having a cryogenically cooled triple resonance TCI probe. 1D $^1H$ NMR spectra were recorded with Bruker standard ZGPR pulse sequence. A total of 128 scans were performed with 64K data points, a relaxation delay of 5 sec, and 16447 Hz spectral width. All the spectra were zero-filled before Fourier Transformation (FT). Following the manual phase and baseline correction, all the free induction decays (FIDs) acquired were processed using the standard Fourier Transformation procedure by Topspin-3.6.1 (Bruker software).

**Cholesterol analysis.** In the current study, we were able to determine the amount of cholesterol in GS samples by simply extracting cholesterol from GS samples into the chloroform solvent. The total quantity of cholesterol was calculated using the integrated area of cholesterol NMR signals and the reference signals. The integral of the cholesterol $^1H$ NMR signals (18-$CH_3$: 0.67 ppm 3-CH: 3.54 ppm & 6-CH: 5.37 ppm) was derived relative to the TSP reference signal (0 ppm).

The concentration of total cholesterol from the $^1H$ spectra can be calculated by integrating the peaks with respect to TSP, and is given by the following formula:

$$\text{Concentration of Total Cholesterol (mmol/g; w/w) } C = \frac{A_c N_t W_t}{A_t M_t N_c W_{gs}} \times 1000 \qquad (1)$$

where $A_c$ and $N_c$ are the integral area and number of protons of the cholesterol components, $W_t = 0.13$mg (weight of TSP in the capillary tube), $A_t = 1$ (integral area of TSP), $M_t = 172.27$ (molecular weight of TSP), $N_t = 9$ (number of protons from TSP). $W_{gs}$ is the weight of GS powder in grams. Through this, the estimated concentration of cholesterol in GS samples is calculated in mmol/g, w/w.

**Solid state NMR experiments.** ~25 mg of GS sample was finely grounded to powder and used to fill the 3.2 mm Zirconia rotor. The $^{13}C$ cross-polarization (CP) magic angle spinning (MAS) NMR experiments were performed at room temperature for the analysis of cholesterol of GS samples. All NMR spectra were acquired on Bruker Biospin Avance III 600 MHz spectrometer, operating at 600.12 MHz for proton $^1H$ resonance frequency and 150.93MHz for carbon $^{13}C$ resonance frequency using Bruker 3.2 mm Magic Angle Spinning (MAS) DVT probe. $^{13}C$ CPMAS spectra were then acquired with the following parameters: Magic angle spinning (MAS) frequency of 5.0 kHz, 256 scans, contact time 1.0 ms, RG value 128, 2k data points, spectral width 301 ppm, and 5 sec recycle delay. $^{13}C$ CPMAS analysis of human GS obtained from patients suffering from malignant (GBC) and benign gallbladder disease (CC)

was performed to identify the different polymorphs of cholesterol. The quantity of cholesterol was also studied using the digital erectic method [26] in the GS obtained from GBC and CC disease.

**Statistical analysis.** We have conducted a one-way analysis of variance (ANOVA) followed by a posthoc test by bonferroni correction using SPSS (IBM USA) software to find out the significant variation in the quantity of cholesterol between groups of different geographical locations and of different gallstone diseases (CC and GBC). The difference between groups included in the study was considered significant when the p-value <0.05.

**Scanning electron microscopy of GS.** SEM analysis was performed on the cleansed pure cholesterol stone, mixed stone, and pigment stone. Each stone was sliced cross-sectional in two pieces (3-5mm), then dried. An electro-conductive adhesive was then used to secure the dried sample on the sample table. The GS was then examined and captured using JEOL SEM, JSM-6490LV.

## Results and discussion

All the gallstones included in the study existed in different colors, shapes, and sizes. In all regions, GS were found in single, double, and multiple numbers. The morphological appearance of the gallstones included in the study was observed carefully by cutting them cross-sectionally. The gallstones included were of all three types; pure cholesterol stones ($\geq$ 70% cholesterol), mixed cholesterol stones (typically 30%-70% cholesterol), and pigment stones ($\leq$30% cholesterol) [27]. In the pure cholesterol stone, cholesterol crystals were arranged radially from the center to the periphery. In the mixed cholesterol stone, cholesterol and pigment layers were arranged alternatively in a crescentic pattern. The pigment stone displays uniformly spread pigment layers. The SEM was performed to show the morphological and structural differences between these gallstones. The SEM analysis revealed that in pure cholesterol stone, the plate and lamella-shaped cholesterol crystals were predominant. In the mixed stone, the distribution of scattered calcium bilirubinate particles was visible along the cholesterol crystals lamella. In the pigment stone, few protein streaks were present along with irregularly distributed calcium bilirubinate particles (Fig 2). Our observations were in the same line with the previous study describing the structural characteristics of the type of gallstones [28]. The [1]H NMR spectra of each type of stone were also acquired and the difference in the concentration of cholesterol was visually revealed. The concentration of cholesterol was higher in the pure cholesterol stone, in the mixed stone, the concentration of cholesterol was less than that of the pure cholesterol stone, while in the pigment stone only traces of cholesterol were observed. The relevant NMR spectra of different types of stones are given in S1 Fig in S1 File.

Out of the total stones collected from various regions, gallstone incidence was found highest in the age groups of 41–50 and 51–60, followed by 31–40 and 61–70 years (Fig 3). The occurrence of GS was found higher in females than in males which was shown in the bar graph. Out of 94 patients, 55 were females and 39 were males. The average age of detection of gallstone disease in India is 51±11 years, compared to 71.2±12.5 years in the Western nations. It was also reported that an increase in age was linked to a higher risk of GBC. Gallstones, age, and sex distribution were found to be consistent with other Indian and international investigations [29–31]. The hormonal variations between males and females may be the cause of this distribution of gallstones. Estrogen replacement therapy and the use of oral contraceptives are the associated risk factors for cholelithiasis [32, 33]. The hepatic bile secretion and gallbladder function is adversely affected by the female sex hormone. Progestins act by reducing bile secretion and affecting gallbladder emptying, resulting in stasis, whereas estrogens increase cholesterol secretion and decrease bile salt secretion [10].

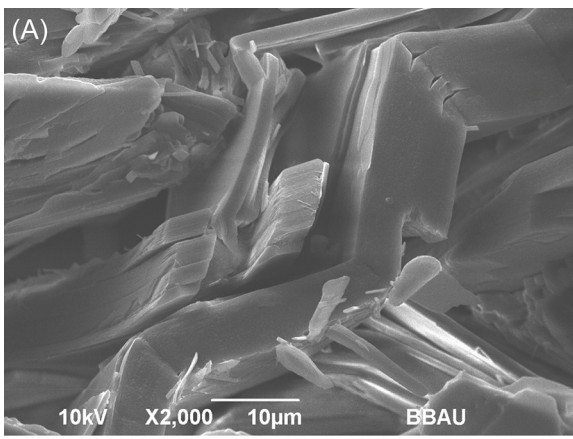

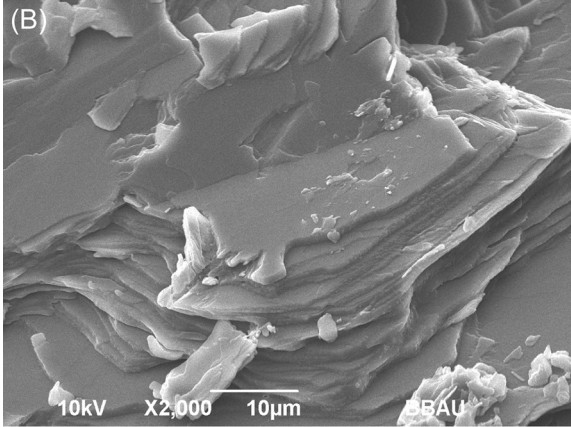

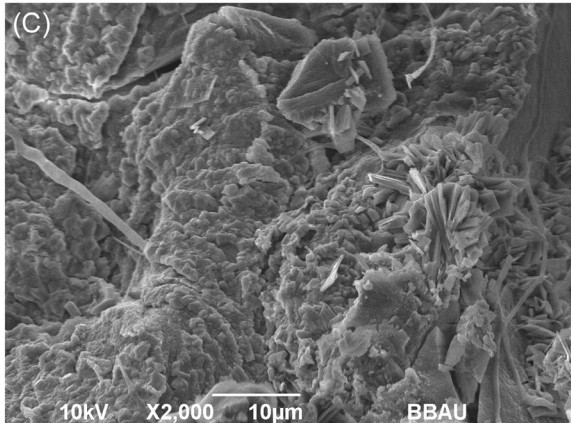

**Fig 2.** The SEM images depicting the microstructure of (A) pure cholesterol stone, (B) mixed stone, and (C) pigment stone.

## Solution state NMR results

In the present study, we analyzed the stones from 5 different areas of northern India, Chandigarh, Delhi, Kangra, Lucknow, and West Bengal, and thereafter compared them with the stones of Southern regions such as Hyderabad and Thiruvananthapuram. We have recorded [1]H NMR spectra from the solution state NMR of all the stones included in the study. Cholesterol was found to be the major component of the maximum stones of Northern India with

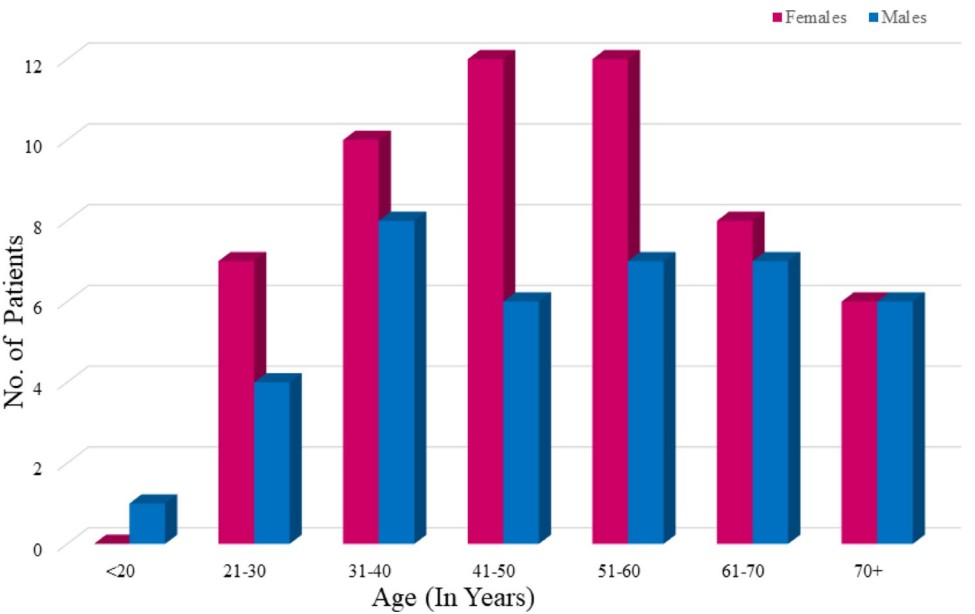

**Fig 3. Frequency of occurrence of gallstones according to age and gender.**

varying concentrations. The stones from South Indian regions had very less cholesterol and mainly contained inorganic salts. Using the reference signal of TSP (0ppm) and $^1$H NMR signals of cholesterol ($18CH_3$: 0.67ppm, 3-CH: 3.54 ppm & 6-CH: 5.37 ppm), the integral area of the cholesterol signal was obtained, and the total amount of cholesterol was calculated (Fig 4). This investigation demonstrated that GS from North India had higher cholesterol levels than GS from South India. In the statistical analysis, we identified significant variations in the concentration of cholesterol amongst North India and South India GS ($p < 0.05$) (S1 Table in S1 File). It is well known that the geographical locations, dietary preferences, and cultural practices of North India and South India are very different from one another. Mustard oil is the preferred choice of cooking oil in the northern and eastern regions of India, while in south India coconut oil and groundnut oil is standard cooking oil [4]. NMR analysis revealed the presence of cholesterol as the major constituent among North Indian GS; however, the South Indian GS showed traces of cholesterol and most GS were pigment stones. The pigment stones were reported to be made up of bilirubin and calcium carbonate. These pigment stones were formed probably due to infection rather than supersaturation of bile [34]. Pigment gallstone formation is also linked with haemolytic disorders [35].

It is unclear why there is such a pronounced difference between the types of gallstones in North India and South India, although this could be related to a genetic predisposition or dietary variations. Diet is likely to play an important role because the amount of calories, fats, and proteins a person consumes affects how much cholesterol is secreted in their bile. Numerous studies have highlighted the connection between food (energy intake, cholesterol, sugar, and carbohydrate) and GS development [24, 36]. Previous studies in which it was reported through analytical characterization that the gallstones from Northern India were mainly cholesterol stones, having a higher concentration of cholesterol when compared with gallstones from Southern India which are mostly pigment stones. Sharma et al. [11] analyzed the gallstones from North India, South India, and UAE, and revealed that cholesterol was a major constituent in the gallstones of North India whereas in South Indian gallstones bilirubin was present predominantly. In another analytical work on the gallstones of North and South India, Ramya

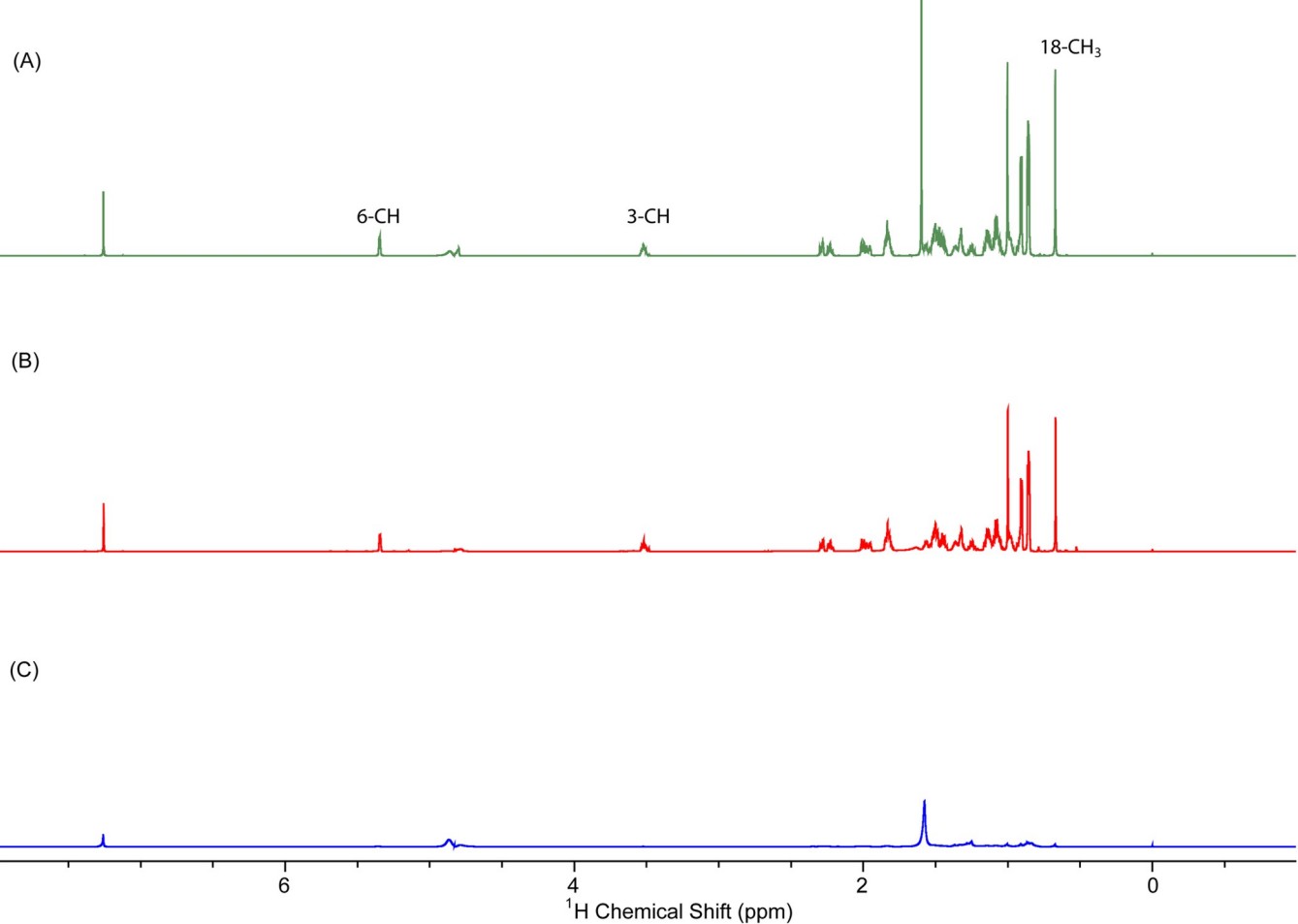

**Fig 4.** [1]H NMR spectra of (A)standard cholesterol, (B) North Indian gallstone, and (C) South Indian gallstone. (18CH$_3$: 0.67ppm, 3-CH: 3.54 ppm & 6-CH: 5.37 ppm peaks depicting the quantity of cholesterol present).

et al. [12] observed that the principal constituent in the North India gallstones was cholesterol. Pichugina et al. [22] not only reported the presence of cholesterol in the gallstones but also provided clues related to the mechanism of formation of gallstones via desmosterol transition. Our findings are in consonance with these studies and highlight the difference in the major component of GS in between North and South India.

## Solid state NMR results

In the biliary tract, one of the most frequent malignancies is gallbladder cancer, which many times supposed to be caused by long-standing gallstone disease. GBC has a very poor prognosis and is frequently detected in an advanced stage [10]. According to reports, GB epithelial metaplastic alterations may progress along an inflammatory and immunological pathway from CC and XGC to GBC [37]. To carry out effective disease management and minimize the mortality rate associated with GBC, understanding the pathogenesis of gallstone disease is important. It is also known that the maximum GBC cases in India are from North India, which is the area with a predominance of cholesterol stones. The link between the CC and GBC conditions needs to be elucidated to halt the development of GBC. Solid state [13]C CPMAS was used for

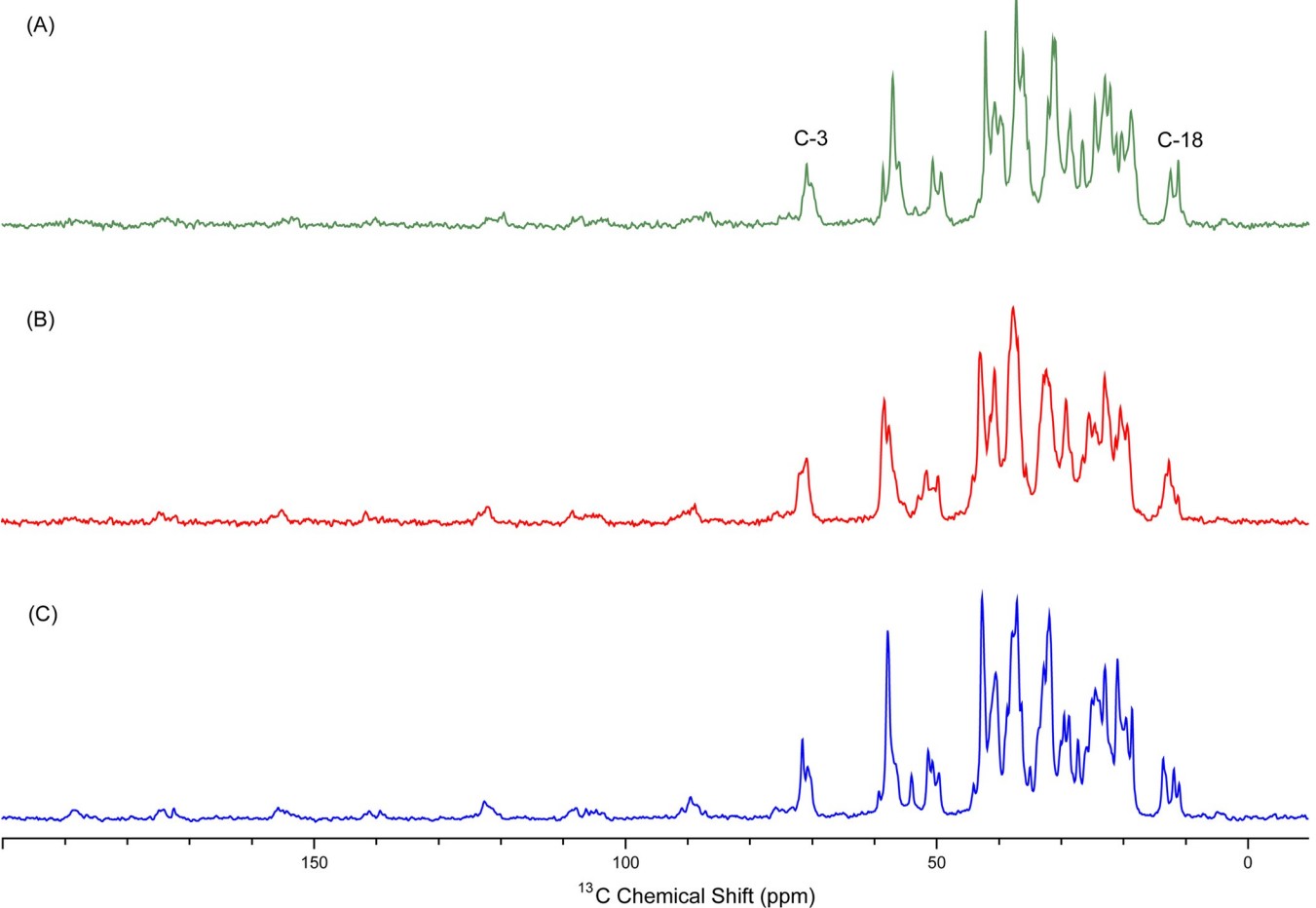

**Fig 5.** $^{13}$C CPMAS NMR spectra of gallstones corresponding to different polymorph forms of cholesterol (A) monohydrate crystalline with an amorphous form and (B) anhydrous form. The difference in the C-18 and C-3 peaks of cholesterol molecules is visible. The spectra (C) correspond to standard cholesterol.

the analysis of stones associated with CC and GBC in the Lucknow and Chandigarh regions. The $^{13}$C CPMAS spectra of the standard cholesterol was also recorded. The spectra obtained from the GS samples showed similarity with the $^{13}$C spectra of cholesterol crystals. The NMR spectra peaks used for the identification and quantification of cholesterol were C-18 (11.0 ppm), and C-3 (70 ppm) (Fig 5). The results of solid state NMR analysis revealed that the difference in the concentration of cholesterol of CC stones of Lucknow from the GBC stones of the same region was significant. Also, the Lucknow CC stones significantly varied in the concentration of cholesterol when compared with the GBC stones of Chandigarh. The level of cholesterol was significantly more in the gallstones samples of benign gallbladder disease (CC) in the Lucknow region compared with GBC stones of Lucknow and Chandigarh. In the same context, Srivastava et al. [21] earlier performed the study on stones of CC and GBC in the Lucknow region only and they reported that the quantity of cholesterol was significantly more in the CC stones when compared with GBC stones of the Lucknow region; this is in the same line as our research, but we have also analyzed the CC and GBC stones of Chandigarh region and when the same has been compared in the view of the quantity of cholesterol, the results varied and there was no significant difference found between the CC and GBC stone of Chandigarh region. So, through our study, we can say that when the region changes the results that

were reported in the earlier study i.e. quantity of cholesterol is higher in CC stones also changes. There is a need to do more extensive research on the stones of CC and GBC from many regions to establish a relationship between GS composition and carcinogenesis.

To highlight the insight into the mechanism of the formation of gallstones, the microstructure of gallstones also needed to be studied. We can detect the atomic level structural variations in the gallstones by observing the $^{13}$C CPMAS spectrum. In the same direction, solid state NMR was used to reveal the crystal polymorphism of cholesterol in the gallstone samples.

$^{13}$C CPMAS experiments of stones of CC and GBC stones were analyzed for identifying the difference in the spectral pattern of CC and GBC stones in two different areas of Northern India namely Chandigarh and Lucknow. On careful observation of spectra of both malignant (GBC) and benign (CC) gallbladder diseases of Lucknow and Chandigarh region, we have found two different types of polymorphs of cholesterol molecules i.e. monohydrate crystalline with amorphous form (ChMA) and anhydrous form (ChA) (Table 1). The C-18 (11.0ppm) and C-3 (70ppm) peaks of the gallstones spectrum help us to conclude the type of cholesterol polymorph. The different form of cholesterol polymorphs have their unique spectral patterns (Fig 6).

Our study revealed that GS of the Lucknow region of both malignant (GBC) (n = 10) and benign gallbladder (CC) (n = 10) diseases belong to the monohydrate crystalline form of cholesterol while GS of Chandigarh region of both malignant and benign gallbladder diseases exists in both monohydrate crystalline form with the amorphous type and anhydrous form (Table 1). Out of 10 CC stones in the Chandigarh region, 5 belonged to monohydrate crystalline with amorphous type and 5 were related to the anhydrous form of cholesterol. Out of 10 GBC stones included in the study, 4 were of monohydrate crystalline with amorphous form and 6 were of anhydrous form. None of the stones included in the solid state NMR study was found to be of monohydrate cholesterol (crystalline) form. Earlier when Jayalaxmi et al. [20] analyzed the type of cholesterol in gallstones from only one region i.e Lucknow, they reported the same as our Lucknow findings that CC stones were of ChMA type while in their study GBC stones were of both monohydrate crystalline and anhydrous form. So, we can state that as we added CC and GBC stones of the Chandigarh region in our study, the previous relationship between the form of cholesterol polymorph and the type of gallbladder disease changed. This indicates that there is a need to study a large number of gallstones of both CC and GBC etiology from different geographical locations to find out the link between the type of cholesterol and type of gallbladder disease (CC & GBC).

The crystallization mechanism of cholesterol has been studied by a few research groups. Konikoff et al. [38] showed that during the crystallization process in model bile, the earliest crystal that form is anhydrous cholesterol. This anhydrous cholesterol is in a filamentous crystals state, which gets gradually covered by a layer of lecithin. These filamentous crystals covered with lecithin work as a substrate for the formation of plates of monohydrate crystals. After some time, transitions of crystals take place into the plate-like cholesterol monohydrate form. In our study, we have also identified that the predominant form of cholesterol in most of the stones was the monohydrate cholesterol form. Apart from this, the involvement of trace

**Table 1. Representing different forms of cholesterol present in gallstones.**

| Region | Disease type | Monohydrate crystalline with amorphous form | Anhydrous cholesterol form |
|---|---|---|---|
| Lucknow | CC (n = 10) | 10 | 0 |
| Lucknow | GBC(n = 10) | 10 | 0 |
| Chandigarh | CC(n = 10) | 5 | 5 |
| Chandigarh | GBC(n = 10) | 4 | 6 |

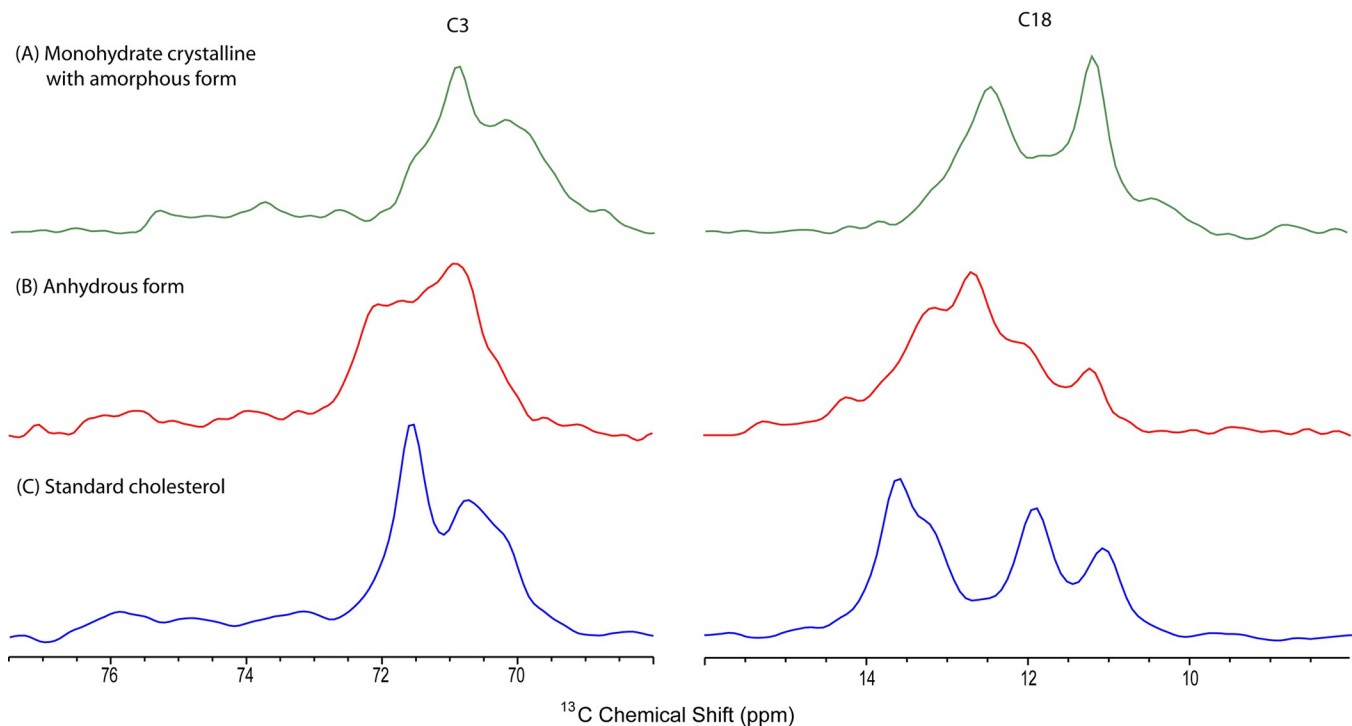

**Fig 6.** [13]C CPMAS NMR spectra of expanded regions of C-18 and C-3 peaks of gallstones cholesterol polymorph highlighting the difference in spectral pattern of these peaks (A) monohydrate crystalline with amorphous form and (B) anhydrous form & (C) corresponds to standard cholesterol.

elements like calcium, iron, zinc, and copper in the formation of gallstones has been defined by some investigations [39, 40]. The changes in serum calcium levels may have an impact on biliary calcium levels, indicating that calcium might play a role in the development of gall-stones. The altered gallbladder motility also found link with low levels of serum calcium, causing biliary stasis and ultimately leading to crystal formation. It has been also demonstrated that serum iron deficiency affects the action of a number of hepatic enzymes, thus increasing bile cholesterol saturation and promoting crystal formation. The inability of the gall bladder mucosa to absorb these metals appears to be the reason of the high concentration of calcium and trace metals (copper & zinc) in the bile of gallstone patients. Hence, the presence of choles-terol crystals may be facilitated by the high biliary cationic concentration. Therefore, to reveal the mechanism of stone formation, it is also crucial to study the role of trace elements in different type of gallstones.

## Conclusion

Studying the composition of gallstones from different geographical regions and different disease conditions such as CC and GBC can give important insight into its formation and correlation with associated diseases. Variations in the gallstone composition between patients with malignant (GBC) and benign (CC) gallbladder disease may offer helpful hints about the aetio-pathogenesis of gallbladder cancer. In the present study, we work to elucidate the composition of gallstones from seven different regions of India, to compare and highlight the major difference, and we have found that the stones from Chandigarh, Delhi, Kangra, Lucknow, and West Bengal which were grouped as North India had cholesterol as a major constituent. In comparison, in the stones of Hyderabad and Thiruvananthapuram, the South India group, only traces

of cholesterol was present and most of them were pigment stones. Along with this, analysis was also performed on the stones of the benign (CC) and cancerous (GBC) gallbladder. This analysis was done to reveal the difference in concentration of cholesterol in the CC and GBC stones of the Lucknow and Chandigarh regions. Also, the type of polymorphs of cholesterol molecules was studied to provide insight into the structure of gallstones. To establish a link between CC and GBC and the type of polymorphs of cholesterol molecules, we need to perform a comprehensive study on a large number of stones including several different geographical locations and various analytical platforms, so that maximum information related to these conditions could be obtained.

## Supporting information

**S1 File. Additional NMR spectra and table of GS analysis.** S1 Fig representing $^1$H NMR spectra of (A) pure cholesterol stone (B) mixed stone and (C) pigment stone depicting the level of cholesterol in different types of stones. S1 Table representing the mean concentration of cholesterol in different regions and the statistical analysis.
(DOCX)

## Acknowledgments

The authors acknowledge the 800MHz & 600 MHz NMR facility at the Centre of Biomedical Research (CBMR), SGPGIMS Campus, Lucknow, and BBAU, Lucknow for providing the SEM facility. The authors also acknowledge Dr. S.K. Mandal for help in the statistical analysis of the study. The manuscript communication number is IU/R&D/2023-MCN000921.

## Author Contributions

**Conceptualization:** Mohammed Haris Siddiqui, V. K. Kapoor, Neeraj Sinha.

**Data curation:** Mohd Adnan Siddiqui, Navneet Dwivedi.

**Formal analysis:** Mohd Adnan Siddiqui, Navneet Dwivedi.

**Funding acquisition:** Neeraj Sinha.

**Investigation:** Mohd Adnan Siddiqui, Navneet Dwivedi.

**Methodology:** V. K. Kapoor.

**Project administration:** Mohammed Haris Siddiqui, Neeraj Sinha.

**Resources:** Mohammed Haris Siddiqui, S. V. Rana, Anil Sharma, N. R. Dash, Rebala Pradeep, Ranjit Vijayahari, Anu Behari, V. K. Kapoor, Neeraj Sinha.

**Software:** Mohd Adnan Siddiqui, Navneet Dwivedi.

**Supervision:** Mohammed Haris Siddiqui, V. K. Kapoor, Neeraj Sinha.

**Validation:** Mohd Adnan Siddiqui, Navneet Dwivedi.

**Visualization:** Mohd Adnan Siddiqui, Navneet Dwivedi.

**Writing – original draft:** Mohd Adnan Siddiqui.

**Writing – review & editing:** Mohd Adnan Siddiqui, Navneet Dwivedi, Mohammed Haris Siddiqui, S. V. Rana, Anil Sharma, N. R. Dash, Rebala Pradeep, Anu Behari, V. K. Kapoor, Neeraj Sinha.

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
