## [Decision Letter · Decision Letter 0]

23 Feb 2023

PONE-D-23-00049NMR SPECTROSCOPY-BASED ANALYSIS OF GALLSTONES OF CANCEROUS AND BENIGN GALLBLADDERS FROM DIFFERENT GEOGRAPHICAL REGIONS OF THE INDIAN SUBCONTINENTPLOS ONE

Dear Dr. Sinha,

Thank you for submitting your manuscript to PLOS ONE. After careful consideration, we feel that it has merit but does not fully meet PLOS ONE’s publication criteria as it currently stands. Therefore, we invite you to submit a revised version of the manuscript that addresses the points raised during the review process.

We look forward to receiving your revised manuscript.

Kind regards,

Thanh-Danh Nguyen, PhD

Academic Editor

PLOS ONE

Journal Requirements:

"NO"

"Acknowledgments: The authors acknowledge the 800MHz & 600 MHz NMR facility at the Centre of Biomedical Research (CBMR), SGPGIMS Campus, Lucknow. The authors acknowledge Dr S.K. Mandal for help in the statistical analysis of the study. M.A.S. acknowledges IUL MCN ………… N.D. acknowledges financial assistance from the Department of Science and Technology (DST), Government of India (IF170726). N.S. acknowledges funding from CBMR, Lucknow."

"NO"

4. We note that Figure 1 in your submission contain copyrighted images. All PLOS content is published under the Creative Commons Attribution License (CC BY 4.0), which means that the manuscript, images, and Supporting Information files will be freely available online, and any third party is permitted to access, download, copy, distribute, and use these materials in any way, even commercially, with proper attribution. For more information, see our copyright guidelines: http://journals.plos.org/plosone/s/licenses-and-copyright.

Reviewers' comments:

Reviewer's Responses to Questions

**Comments to the Author**

1. Is the manuscript technically sound, and do the data support the conclusions?

Reviewer #1: Yes

Reviewer #2: Yes

2. Has the statistical analysis been performed appropriately and rigorously? 

Reviewer #1: No

Reviewer #2: I Don't Know

3. Have the authors made all data underlying the findings in their manuscript fully available?

Reviewer #1: Yes

Reviewer #2: Yes

4. Is the manuscript presented in an intelligible fashion and written in standard English?

Reviewer #1: Yes

Reviewer #2: Yes

5. Review Comments to the Author

Reviewer #1: Overall, the authors have got the NRM spectrum from patient gallstones from various regions in India. The results have shown the different in Gallstones components in NMR.

There are some recommends and questions for the authors.

Recommendation:

It is not necessary to have figure 1 and 2 in the manuscript.

Authors could add Indian map. On the map, authors provide the location of collected samples, number, samples, sexuality, type/color/image of gallstones.

Author should have the period of collecting samples in Materials and methods

Question:

1. In line 223-224, Authors have pointed out the mis-classification by morphology of gallstone. This may happen. So do the author compare between the NMR spectrum and morphology/ color/type of gallstone in the study? How many precents of correction when diagnosis by looking at the morphology/color of gallstones by Doctor?

2. Authors have found many chemical in gallstones. Do the authors take samples, analyze and confirm the chemical in the gallstone by another methods?

3. Identifying type of gallstone is very important, sometimes, patients don’t need to remove the gallbladder or gall track. So is it possible to perform the Magnetic resonance spectrum (MRS) or MRI in patients to find out the component of gallstones? Authors may consider the time, magnetic force for scanning.

Reviewer #2: The manuscript by Siddiqui et al. describes the characterization of gallstones from India using NMR spectroscopy.

Major changes:

pg. 138-139Authors should provide SEM images of cross-sections of all the studied gallstones. Optical images could be provided to show the overall morphological differences.

pg.313-315 It's not clear if Figure 6's spectra correspond to the gallstones from CC and GBC patients. Please rephrase the corresponding text and Figure caption. Add labels in Figure 6.

pg. 343-350 Please discuss the effect of trace elements during the crystallization process.

Minor changes:

pg. 24 Explain GS abbreviation in the abstract and in its first appearance in the main text.

pg. 85 Leave space between scanning electron microscopy and (SEM)

pg. 91-99 Discuss the work in the previous regions of India about gallstones

pg. 105 Lowercase Solid-State. Apply to all text.

pg. 108 Lowercase Solution-State. Apply to all text.

pg.163 Superscript 1 in 1H NMR. Apply to all text.

pg.203-205 Could you provide percentages of the cholesterol

pg. 223-231 This information should be mentioned earlier in the introduction

pg. 240-242 Please refer to the specific differences between the regions and add references

pg. 247 What do you mean by molecules? Calcium carbonate is a inorganic structure. You could add mineral.

pg. 373 Complete the missing information

6. PLOS authors have the option to publish the peer review history of their article (what does this mean?). If published, this will include your full peer review and any attached files.

Reviewer #1: **Yes: **Hieu Vu-Quang

Reviewer #2: No

---

## [Author Response · Author response to Decision Letter 0]

18 May 2023

Editor Comments

Response: We have followed PLOS ONE's style guidelines for the preparation of the revised manuscript.

"NO"

Response: The authors received no specific funding for this work. The clarification statement has been made in the cover letter also.

"Acknowledgments: The authors acknowledge the 800MHz & 600 MHz NMR facility at the Centre of Biomedical Research (CBMR), SGPGIMS Campus, Lucknow. The authors acknowledge Dr S.K. Mandal for help in the statistical analysis of the study. M.A.S. acknowledges IUL MCN ………… N.D. acknowledges financial assistance from the Department of Science and Technology (DST), Government of India (IF170726). N.S. acknowledges funding from CBMR, Lucknow."

"NO"

Response: The authors received no specific funding for this work and required changes has been made in the “Acknowledgements section”.

4. We note that Figure 1 in your submission contain copyrighted images. All PLOS content is published under the Creative Commons Attribution License (CC BY 4.0), which means that the manuscript, images, and Supporting Information files will be freely available online, and any third party is permitted to access, download, copy, distribute, and use these materials in any way, even commercially, with proper attribution. For more information, see our copyright guidelines: http://journals.plos.org/plosone/s/licenses-and-copyright.

Response: The Figure 1 from has been replaced with the new Figure (Indian Map) in the revised manuscript, as suggested by Reviewer #1.

5. Please include captions for your Supporting Information files at the end of your manuscript, and update any in-text citations to match accordingly.

Response: We have included the text at the end of manuscript as per PLOS ONE's supporting information guidelines.

Reviewer #1: 

Recommendation:

1. It is not necessary to have figure 1 and 2 in the manuscript.

Response: The suggestion is appreciated and we have removed the Figure 1 and Figure 2.

2. Authors could add Indian map. On the map, authors provide the location of collected samples, number, samples, sexuality, type/color/image of gallstones.

Response: Thanks for providing valuable suggestion. We have incorporated the Indian map in the revised the manuscript on page no. 5. 

3. Author should have the period of collecting samples in Materials and methods.

Response: The suggestion is acknowledged and the period of collecting the samples has been mentioned in the revised manuscript line no. 163-164.

Questions:

1. In line 223-224, Authors have pointed out the mis-classification by morphology of gallstone. This may happen. So do the author compare between the NMR spectrum and morphology/ color/type of gallstone in the study? How many precents of correction when diagnosis by looking at the morphology/color of gallstones by Doctor?

Response: We agree with the reviewer comments. NMR spectrum and its comparison with type of stone is presented in SI Figure No 1. The diagnosis of gallstones diseases based solely on the morphology and color of the stones can be unreliable. The appearance of gallstones can provide some clues as to their composition. The accuracy of diagnosis depends greatly on the experience level and expertise of the doctor performing the diagnosis.

2. Authors have found many chemical in gallstones. Do the authors take samples, analyze and confirm the chemical in the gallstone by another methods?

Response: NMR spectroscopy is the chosen method, and it is a powerful technique for the characterization of compounds, including those present in gallstones. We have utilized both, the solution-state NMR and solid-state NMR technique to provide comprehensive insight of the major component of the gallstones. Along with this, we have used scanning electron microscopy (SEM) to reveal the morphological and structural differences between the different types of stone included in the study i.e. cholesterol stone, mixed stone and pigment stone. The SEM images showing the plate like cholesterol crystals with laminar and radial arrangement and confirmed the presence of cholesterol in the gallstones. Our results are in the same line with previous studies (reference no 18 in the revised manuscript). 

3. Identifying type of gallstone is very important, sometimes, patients don’t need to remove the gallbladder or gall track. So is it possible to perform the Magnetic resonance spectrum (MRS) or MRI in patients to find out the component of gallstones? Authors may consider the time, magnetic force for scanning.

Response: Yes, it is possible to use Magnetic Resonance Imaging (MRI) or Magnetic Resonance Spectroscopy (MRS) to identify the type of gallstone. MRI can provide high-resolution images of the gallbladder and gallstones, allowing clinicians to determine the composition of the stone. One of the study that uses the MRI for the analysis of different compositions of gallstones shows that on T1-weighted 3D fast spoiled gradient-echo images, of the pigment gallstones were hyper-intense and the cholesterol gallstones were hypo-intense. So, we can say that based on the differences of signal intensity of gallstones, the 3D fast spoiled gradient-echo T1-weighted imaging is able to diagnose the composition of gallstones (reference no.19 in revised manuscript). However, it is important to note that not all hospitals and medical facilities may have the equipment and expertise to perform MRS. Additionally, MRI and MRS scans can be time-consuming and may not be suitable for all patients, especially those with certain medical conditions (with pacemaker) or who cannot tolerate the magnetic force used in the scans.

Reviewer #2: 

1. Pg. 138-139 Authors should provide SEM images of cross-sections of all the studied gallstones. Optical images could be provided to show the overall morphological differences.

Response: We appreciate the suggestion to add SEM images and optical images to provide further insight into the morphological differences between the gallstones studied. Mainly three types of gallstones from different geographical regions and etiology have been analyzed which are cholesterol stone, mixed stone, and pigment stone. To show the morphological and structural differences between them the SEM analysis of these gallstones have been included in the revised manuscript on page 9-10.

2. pg.313-315 It's not clear if Figure 6's spectra correspond to the gallstones from CC and GBC patients. Please rephrase the corresponding text and Figure caption. Add labels in Figure 6.

Response: Thank you for your valuable suggestion. Figure 6 shows the type of polymorphs of cholesterol (monohydrate crystalline with amorphous form and anhydrous form) which can be present in either CC or GBC gallstones, that can also be understood by Table 1. No specific form of cholesterol is associated with either CC or GBC gallstones. Figure caption of Figure 6 has been reframed for clear understanding on line no. 353-356.

3. pg. 343-350 please discuss the effect of trace elements during the crystallization process.

Response: We appreciate the suggestion and the role of trace elements in the formation of gallstones is incorporated in the manuscript in line no. 383-395.

4. pg. 24 Explain GS abbreviation in the abstract and in its first appearance in the main text.

Response: Thanks for highlighting this. The GS abbreviation is now mentioned in the abstract section page no. 2 line no. 44.

5. pg. 85 Leave space between scanning electron microscopy and (SEM).

Response: The suggested correction has been made in the revised manuscript line no. 110.

6. pg. 91-99 Discuss the work in the previous regions of India about gallstones.

Response: Thanks for the suggestion. The work from previous studies have been incorporated on page no. 4 line no. 104-106 and page no. 12 line no. 297-302.

7. pg. 105 Lowercase Solid-State. Apply to all text.

Response: The suggestion is appreciated and changes has been incorporated in the revised manuscript.

8. pg. 108 Lowercase Solution-State. Apply to all text.

Response: The suggestion is appreciated and changes has been incorporated in the revised manuscript.

9. pg.163 Superscript 1 in 1H NMR. Apply to all text.

Response: The suggestion is appreciated and changes has been incorporated in the revised manuscript.

10. pg.203-205 Could you provide percentages of the cholesterol.

Response: The percentage of cholesterol in each type of gallstone varies: pure cholesterol stones: (≥70% cholesterol), mixed stones: (30-70% cholesterol), and pigment stones: (≤30% cholesterol). This has been incorporated in line no. 231-233.

11. pg. 223-231 This information should be mentioned earlier in the introduction.

Response: The suggestion is appreciated and the information has been mentioned in the introduction section on page no. 4 line no. 100-104 of the revised manuscript.

12. pg. 240-242 Please refer to the specific differences between the regions and add references.

Response: Thanks for the suggestion. The suggestion has been incorporated on page no. 11 line no. 278-280.

13. pg. 247 What do you mean by molecules? Calcium carbonate is a inorganic structure. You could add mineral.

Response: Authors would like to apologize for the mistake. The appropriate changes has been made in the revised manuscript. 

14. pg. 373 Complete the missing information.

Response: The suggestions have been incorporated in the acknowledgement section.

---

## [Editor Report · Decision Letter 1]

29 May 2023

NMR spectroscopy-based analysis of gallstones of cancerous and benign gallbladders from different geographical regions of the Indian subcontinent

PONE-D-23-00049R1

Dear Dr. Sinha,

We’re pleased to inform you that your manuscript has been judged scientifically suitable for publication and will be formally accepted for publication once it meets all outstanding technical requirements.

Kind regards,

Thanh-Danh Nguyen, PhD

Academic Editor

PLOS ONE
---

## [Editor Report · Acceptance letter]

14 Jun 2023

PONE-D-23-00049R1 

NMR spectroscopy-based analysis of gallstones of cancerous and benign gallbladders from different geographical regions of the Indian subcontinent 

Dear Dr. Sinha:

I'm pleased to inform you that your manuscript has been deemed suitable for publication in PLOS ONE. Congratulations! Your manuscript is now with our production department. 

Kind regards, 

on behalf of

Dr. Thanh-Danh Nguyen 

Academic Editor

PLOS ONE